# Selective Electromagnetic Measurements of 4G Signals: Results of an Italian National Intercomparison

**Lucia Ardoino [1,]\*, Sara Adda [2], Laura Anglesio [2] and Enrichetta Barbieri [3]**

[1]    ENEA—Territorial and Production Systems Sustainability Department, Technologies and methods for the protection of health (SSPT-TECS_SAM), via Anguillarese, 00123 Rome, Italy

[2]    Regional Agency for the Protection of the Environment of Piedmont (ARPA Piemonte), Physical and Technological Risks Department, Via Jervis, 30-10015 Ivrea (TO), Italy; sara.adda@arpa.piemonte.it (S.A.); laura.anglesio@arpa.piemonte.it (L.A.)

[3]    ISPRA—National Center for environmental characterization and protection of the coastal strip and operational oceanography (CN-COS), Via del Cedro (c/o Dogana d'Acqua), 57122 Livorno, Italy; enrichetta.barbieri@isprambiente.it

\*    Correspondence: lucia.ardoino@enea.it

**Abstract:** In June 2016, with the aim of ensuring a global improvement in the performance of the Italian System of the Environmental Agencies (SNPA) and its homogeneity on the national territory, an intercomparison circuit (IC) was planned and conducted concerning the measurements of electromagnetic fields associated with Long Term Evolution (LTE) mobile communications, which were very recently introduced at that time. The intercomparison circuit, designed and built according to the criteria of ISO 17043, was organized as part of a consolidated collaboration between the Institute for Environmental Protection and Research (ISPRA) and the Piedmont Regional Agency for Environmental Protection (Arpa Piemonte). The results obtained, preceded by a brief description of the entire process of organization and analysis, are the subject of this work. The IC covered in particular: the narrow band measurement procedures used in the field; the choice of decoding, measurement and extrapolation of the synthesis result; the response of the instrumentation, limited to the models in the field. The site chosen by the organizers, primarily characterized through measurements and theoretical evaluation of the field, is the roof of the Lingotto Building in Turin. A total of 27 groups participated in the circuit: 25 SNPA departments (including the organizers Arpa Piemonte and ISPRA) and 2 private labs. All participants provided the results. The outcome of the comparison was decidedly positive: only 2 participants, for whom a joint assessment of possible causes will also be illustrated (according to ISO 5725:2), achieved significantly different results.

**Keywords:** electromagnetic field measurements; LTE signal; inter-laboratory comparisons; proficiency test for physical measurements

---

## 1. Introduction

Participation in inter-laboratory comparison programs is now a key element of any accreditation process and is generally required in the context of quality assurance of laboratory results.

The main purpose of intercomparison circuits (also interlaboratory comparison) is to provide participants with objective tools for demonstrating the reliability of their measurement results, by implementing a common protocol and compared to the results provided by the other participants. This is particularly relevant in situations, such as measurements of environmental electromagnetic field levels, where measurements are carried out directly in the field and not in laboratory and where the

quantities to be measured (measurands) are physical magnitudes associated with signals intrinsically variable over time [1].

In addition, in the context of mobile communication and individual services, the "platforms used" for transmissions are constantly and rapidly transforming, with changes in the physical signal (electromagnetic field) and in the network architecture.

As a result, it would always be desirable to rely on a large number (>15) of experienced laboratories to participate in an intercomparison circuit. The National Environmental Protection System (SNPA) is therefore the ideal ground for such processes.

For these reasons, in accordance with the need for the agencies to verify the reliability of their results (especially on a new signal) and with the coordinating role of the Institute for Environmental Protection and Research (ISPRA), the intercomparison circuit IC_ISPRA2016_LTE was planned and conducted. The intercomparison (IC) was about Long Term Evolution (LTE) signal measurements.

In the specific case of measurements of environmental electromagnetic field levels associated with transmissions and telecommunications of complex signals, such as LTE, two different aspects can be defined whose final assessment requires to be a topic of intercomparison:

- the response of the instrumentation, and
- the measurement procedures used.

    Both items include several "sub-items".
    The IC_ISPRA2016_LTE covered:

- Narrow band measurement procedures used in the field, which is an issue of particular importance due to the recent (in 2016) introduction of LTE signals, and the resulting lack of detailed measurement procedures and minimal experience of agency operators with low traffic signals;
- The choice of how to decode, measure and extrapolate the synthesis result, since such choices, in the case of signals with complex digital modulation, heavily affect the final result of the measurement.

## 2. Materials and Methods

The organization of an intercomparison circuit of this type requires, first, the identification and characterization of a suitable site [2,3]. In this case, thanks to the land registry of the Arpa Piemonte and to the fact that it was a recently introduced source, a potential site was identified in the roof of the Lingotto building, where, on two opposite sides, two antennas are placed with carriers in different bands. The characterization required two measurement campaigns carried out through a period of several months.

Once the appropriateness of the site was verified, a preliminary protocol was established with information relating to the purposes of the intercomparison, dates and places where it would take place and the minimum requirements for participation, in terms of personnel, tools and procedures.

As in a previous intercomparison circuit ([4]), the requirements for participation were simply those of having, at the time of the implementation of the field tests, properly trained personnel, and having adequate approved instruments. Of course, all the instruments had to be provided with a calibration certificate issued by an Accredited Calibration Laboratories (LAT "*Laboratorio Accreditato di Taratura*", certificate in Italy) on which the reference chain and uncertainty were reported. Decoding tools were not required because, at the time of the measurements in the field, many laboratories did not have these options available on the instrument.

Participants were asked to adhere to the dictates of the final protocol that was sent to them at the end of registration, regarding shifts and positions on the site, and with regard to certain details on how to acquire and process data (according the technical norm CEI 211-7 App. E, [3]).

All participants provided their results in accordance with the protocol: they had to: fill in a predefined Excel sheet with raw and processed data (results) and relative uncertainty for each measurement session, and provide the technical report, drawn up according to their custom, in which it was possible to report any data deemed "unreliable."

In summary, the organization of the IC was managed according the following steps:

- Site characterization and analysis of the results of preliminary measurements, in order to identify sufficiently homogeneous areas,
- Preparation of a pattern of measurement points with the possibility of repetition,
- Organization and logistics management (access permissions, electrical outlets, etc.),
- Measurements on site,
- Collection and analysis of results,
- Preparation and submission of the final report and the Individual Evaluation Reports for the participants.

### 2.1. Site Characterization

The chosen site is the top floor, "the track", of the Lingotto building in Turin where, at the two curves, two LTE signal antennas are installed at 800 and 1800 MHz belonging, respectively, to TIM and Vodafone. The area used for the intercomparison circuit was only that of the north side, covered by the signal of Telecom Italy (TIM) operator at 800 MHz (Figure 1), which was the chosen band.

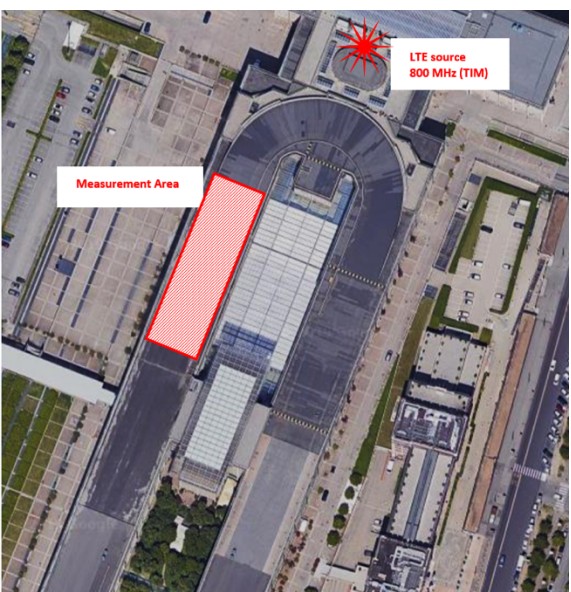

**Figure 1.** The measurement site (sample site) seen from above (photo copyright: Agricultural disbursement agency).

To verify the suitability of the site for use as a "sample site" for an intercomparison, several measurement sessions on the site were carried out in far field area (about 175λ) and compared with numerical evaluation.

Measurements for characterization were carried out in an area of about 14 × 18 m, in a grid of 2.5 m pitch points for a total of 42 measurement points. Measurements were performed both in Channel Power (CP) mode and Reference Signals (RS) measurements; for both measurement modalities, the uniformity of field levels was assessed through the dispersion of the values, with a variation coefficient of 16% (dispersion obtained from the Standard Deviation of the measurement set in the worst case of the entire area). The analysis of the spatial distribution of the field (Figure 2) revealed a good level of uniformity with the only exclusion of some critical points (points closer to the source or in proximity of metal objects). The signal level was low but detectable. The area extension allowed selecting a set of points in which to allocate at least fifteen laboratories at the same time (Figure 3).

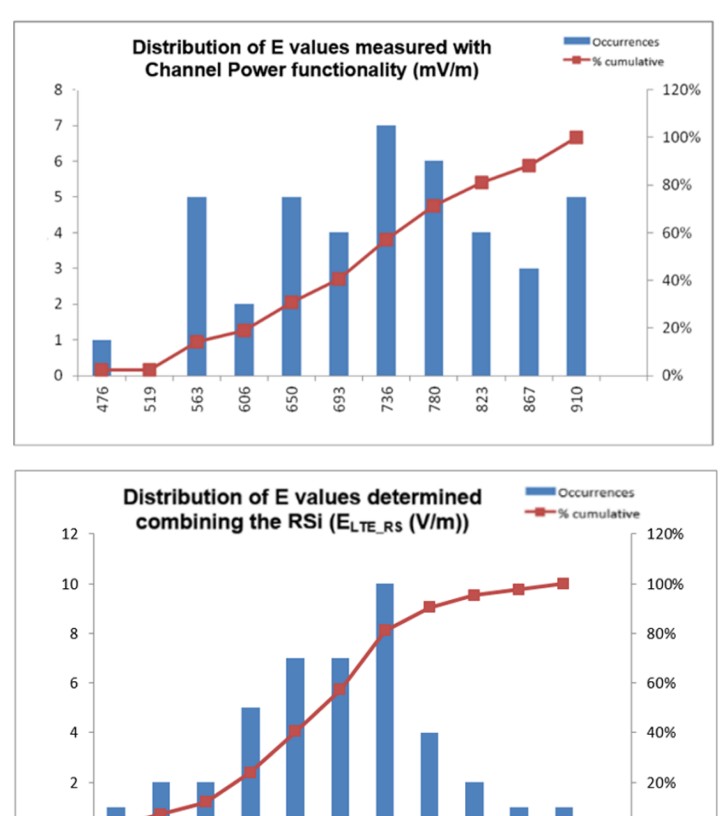

**Figure 2.** E value distributions measured in the RSi and Channel Power (CP) modalities. The distributions obtained had a variation coefficient of 16% and 14%.

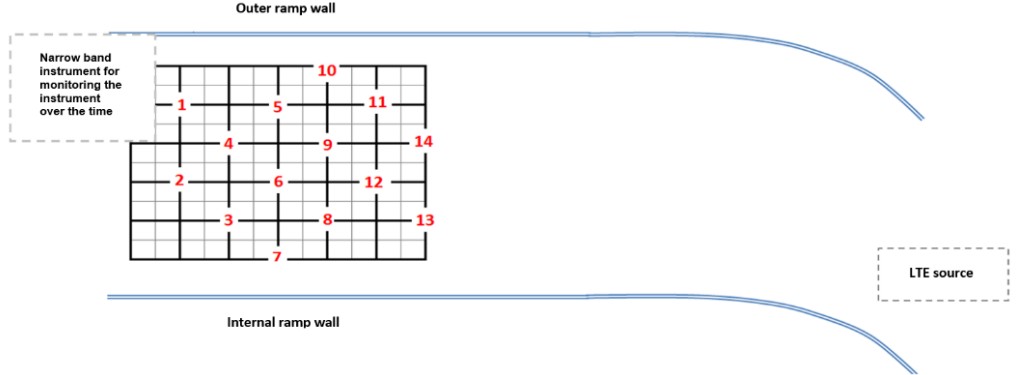

**Figure 3.** The scheme of the positions in the site (not in scale).

Due to the recent introduction of this signal (at the time of the first measurements, 2015) and, therefore, the reduced amount of traffic, the characterization measurements were repeated after about six months (measurements 2016), close to the date of the intercomparison, to verify that the signal level remained homogeneous and detectable.

Both measurement sessions were conducted with a spectrum analyzer Narda SRM3006 (Narda STS Italy – Cisano sul Neva), provided with the predefined routine for LTE – Frequency Division Duplexing (FDD)/Time Division Duplexing (TDD) signals measurements and its isotropic antenna (75 MHz–3 GHz). The Channel power measurements were performed with the appropriate parameters ($f_c$ = 806 MHz, Integration Bandwidth = 10 MHz, Resolution Bandwidth = 100 kHz, trace average

over 6 min); RS measurements were performed by setting the instrument to LTE standard tool with a decoding band of 10 MHz, that selects automatically measuring parameters.

The results of the characterization measurements are partially shown in the section on the results, together with those of the analysis of the participants' measurements.

### 2.2. Inter-Comparison Measurement Plan

The program of the measurement on site included two measurement sessions, in two separate days, of six rounds each. Each participant made measurements in a single session, in the six predefined positions, one for each turn, according to the sequence shown in Table 1 (the positions are represented in Figure 3). The sample of the six measurements taken by each participant in the sequence of the six positions will be referred to in the following as a "sestet".

**Table 1.** Sequences of positions, according the previous scheme, for each participant ("sestet").

| Session 1 | Session 2 | Code for the Sequence of Measurement Points ("Sestet") | Turn | | | | | |
|---|---|---|---|---|---|---|---|---|
| Participant Codes | | | 1° | 2° | 3° | 4° | 5° | 6° |
| A01 | A02 | A | 1 | 5 | 11 | 13 | 6 | 10 |
| A20 | A04 | B | 2 | 6 | 12 | 14 | 10 | 5 |
| B05 | A06 | C | 3 | 8 | 13 | 11 | 5 | 1 |
| A07 | A08 | D | 4 | 9 | 14 | 12 | 7 | 2 |
| B09 | B27 | E | 5 | 11 | 8 | 3 | 2 | 13 |
| B11 | B15 | F | 6 | 12 | 10 | 4 | 1 | 14 |
| B13 | A14 | G | 7 | 2 | 4 | 10 | 11 | 8 |
| A12 | A16 | H | 8 | 13 | 5 | 9 | 3 | 12 |
| A17 | B18 | I | 9 | 14 | 7 | 2 | 4 | 11 |
| B26 | - | L | 10 | 1 | 3 | 6 | 12 | 7 |
| B03 | B21 | M | 11 | 4 | 2 | 7 | 13 | 9 |
| A22 | A24 | N | 12 | 7 | 9 | 8 | 14 | 6 |
| A25 | A19 | O | 13 | 3 | 6 | 1 | 9 | 4 |
| A10 | A28 | P | 14 | 10 | 1 | 5 | 8 | 3 |

The choice of the positions of each sequence followed some specific principle: each participant had to take measurements in all sectors of the area, without ever repeating the measurement at the same point and without two participants taking measurements in the same round in the same position.

During each turn, participants took measurements in their assigned position within 30 min, during which they acquired the electric field values in all the modes covered (Channel Power, Decoding RSi and SpanZero) by the Protocol and in accordance with CEI 211-7 App. E.

All participants were asked to provide the results of the electric field measurement in CP mode while the measurement of the Reference Signals (and/or the extrapolated electric field) was asked to all groups able to decode the signal (groups provided by the instruments able to decode). The SpanZero mode measurement was optional for everyone.

Only 3 participants did not return the results of the CP measurement (therefore, it was carried out by 24 groups), all groups with decoding tools provided the RSi values (19) and 12 participants returned the results of the measurement in SpanZero mode. Eight participants took measurements in each of the three modes.

During all measurement sessions, to ensure that there were no significant changes in the level of the interest signals, a "tool" for continuous narrow band capture was placed near the measurement area. The instrument used for this purpose was the same Narda SRM3006 used for the preliminary characterization measurements sets for the acquisition of the trend over time of $E_{CP}$ (Channel Power mode), which is more representative for the amplitude variation of the signal due to the traffic. The maximum variation of channel power levels in half an hour was below 8% of the mean value.

### 2.3. Analysis of Data and Results

The analysis of the results was preceded by a "numeric verification" phase aimed mainly at detecting any transcription errors and carried out by comparing the data provided with the

predefined Excel sheets and those reported in the technical reports. The electric field $E_{LTE}$ values of the extrapolations have always been recalculated to verify the correct application of the standard.

In the entire analysis procedure, only results declared "unreliable" by the participant were excluded. No corrections or exclusions of any kind were made, not even the few outliers present in some samples.

For each measurement mode, the value to be assigned to the measurement was determined as a consensus value from the participants' results, after compared to the value determined during characterization. This consensus value, determined by robust statistics (Annex C of ISO13528, Algorithm A), is representative of the situation at the time of the measurements.

The results were analyzed according to two different schemes called "by positions" and "by sestet" respectively.

In the "by position" analysis, a "sample" was created for each of the 14 positions with the results of all participants who measured in that position in the different rounds of the two measurement sessions: each sample consists of a maximum of 12 values (Table 2).

**Table 2.** Measurement samples in CP mode for all positions with their average values, standard deviation, and uncertainty.

| Positions Session/Turn | 1 | 2 | 3 | 4 | 5 | 6 | 7 | 8 | 9 | 10 | 11 | 12 | 13 | 14 |
|---|---|---|---|---|---|---|---|---|---|---|---|---|---|---|
| **1-1** | 0.7 | 0.694 | 1.22 | 0.27 | 2.72 | 0.26 | | | 0.96 | 0.738 | 0.59 | 0.89 | 0.68 | 0.393 |
| **1-2** | 1.038 | | 0.976 | 0.57 | 0.75 | 0.956 | 1.46 | 2.1 | 0.62 | 0.634 | | 0.51 | | 1.18 |
| **1-3** | 0.351 | 0.71 | 0.832 | | | 0.726 | 0.86 | 0.7079 | 0.8 | 0.35 | 0.68 | 0.814 | 1.24 | 0.36 |
| **1-4** | 0.642 | 0.65 | 0.3736 | 0.31 | 0.358 | 0.791 | 0.99 | 0.72 | | | 1.49 | 0.4 | 0.59 | 0.762 |
| **1-5** | 0.27 | 0.457 | | 0.7 | 2.22 | 0.72 | 0.56 | 0.173 | 0.966 | 0.916 | | 0.998 | 0.77 | 1.02 |
| **1-6** | 1.23 | 0.32 | 0.407 | 0.568 | 0.812 | 0.76 | 0.777 | | 0.74 | 0.72 | 0.95 | | 0.3618 | 0.32 |
| **2-1** | 0.61 | 0.6 | 0.87 | 0.518 | 0.5 | 0.66 | 0.1 | | 0.74 | x | 0.61 | 2 | 0.777 | 1.08 |
| **2-2** | | 0.33 | 0.788 | 0.7 | 0.84 | 0.7 | 2 | 0.76 | 0.85 | 0.76 | 0.45 | 0.81 | | 0.65 |
| **2-3** | 1.95 | 0.77 | | 0.31 | | 1.037 | 1.06 | 0.6 | 2.1 | 0.9 | 0.95 | 1 | 0.91 | 1.15 |
| **2-4** | 0.808 | 0.83 | 0.54 | 0.48 | 0.81 | | | 2 | | 0.45 | 0.74 | 0.735 | 0.77 | 0.9 |
| **2-5** | 0.77 | 0.4 | | 0.46 | 0.93 | 0.92 | 1.155 | 0.5 | 1.07 | 0.9 | 0.44 | | 1.03 | 1.2 |
| **2-6** | 0.79 | 0.556 | 0.56 | 0.734 | 0.8 | 1.3 | x | 0.37 | | 0.83 | 1.07 | | 0.4 | 0.59 |
| Robust Avg | **0.77** | **0.60** | **0.79** | **0.52** | **0.82** | **0.79** | **0.98** | **0.69** | **0.87** | **0.75** | **0.71** | **0.82** | **0.77** | **0.83** |
| Rob StDev | 0.28 | 0.21 | 0.34 | 0.27 | 0.16 | 0.17 | 0.39 | 0.36 | 0.19 | 0.20 | 0.36 | 0.29 | 0.24 | 0.42 |
| Uncertainty | **0.11** | **0.08** | **0.14** | **0.10** | **0.06** | **0.06** | **0.15** | **0.15** | **0.08** | **0.08** | **0.14** | **0.12** | **0.09** | **0.15** |

Each sample was evaluated for normal distribution of data (using Normal Probability Plot) and for the presence of outliers (Huber tests). Robust statistics was applied to determine the consensus value to assign to the "measurand" instead of the "reference value", the standard deviation and the uncertainty of this value, used both to represent the set of sample values by PomPlot and to determine "partial" z-scores (i.e., related to the single measurement).

The same procedure was used for both results of Channel Power ($E_{CP}$) measurements and those of measurements of values extrapolated from RS ($E_{LTE\_RS}$) (Table 3). For SpanZero measurements, instead, the small number of results provided did not make it possible to apply the "positional" analysis to those measurements.

In the "by sestet" analysis, the six results of the measurements taken by each participant in the six assigned positions were treated as "repetitions of the measurement at different points in a uniform area" from the point of view of the electric field level. The average of the measurements was therefore determined as a representative value of the participant's measurement (the measurement of the sestet).

The sample of these sestets, one for each participant, was then subjected to robust statistics to determine the consensus value to be assigned to the measurement, and the related standard deviation and uncertainty.

**Table 3.** Measurement samples in $E_{LTE\_RS}$ mode for all positions with their average values, standard deviation, and uncertainty.

| Positions Session/Turn | 1 | 2 | 3 | 4 | 5 | 6 | 7 | 8 | 9 | 10 | 11 | 12 | 13 | 14 |
|---|---|---|---|---|---|---|---|---|---|---|---|---|---|---|
| 1-1 | 1.888 | - | - | - | - | 0.883 | 1.860 | 1.200 | - | 1.892 | - | 2.289 | 1.794 | 1.023 |
| 1-2 | 1.752 | 1.150 | 2.102 | 1.300 | 1.784 | - | 2.113 | - | - | 0.387 | - | 1.225 | 1.400 | - |
| 1-3 | 0.850 | 1.900 | 2.131 | 0.930 | 1.800 | 2.201 | - | 1.475 | 1.955 | 1.225 | 1.906 | - | - | - |
| 1-4 | 1.525 | - | 1.190 | 1.039 | 0.850 | 1.871 | 1.900 | 1.941 | 2.000 | 2.100 | - | - | 1.802 | - |
| 1-5 | 0.693 | 1.122 | 1.400 | - | - | 1.715 | - | 0.364 | 2.096 | - | 1.760 | 2.131 | 1.500 | 1.957 |
| 1-6 | - | - | 1.196 | 1.408 | - | 1.703 | 1.848 | 1.670 | 1.900 | 1.756 | - | 1.400 | 0.991 | 1.039 |
| 2-1 | 1.249 | 1.855 | 1.355 | 1.568 | - | 1.646 | 1.907 | - | 2.364 | - | - | 1.386 | 2.063 | - |
| 2-2 | - | 1.040 | 1.931 | - | 3.118 | 1.854 | 1.386 | 1.230 | 2.165 | - | - | 2.081 | - | 2.536 |
| 2-3 | - | - | - | 0.745 | - | 1.665 | 2.582 | - | 1.732 | 1.837 | 2.258 | 2.152 | 1.387 | 2.358 |
| 2-4 | 1.665 | 2.330 | - | 1.438 | - | - | - | 1.428 | - | 1.353 | 0.608 | 1.579 | 2.020 | 1.945 |
| 2-5 | 1.907 | - | - | 1.632 | 1.435 | 1.549 | 2.581 | - | 1.939 | 1.873 | 1.249 | - | - | 0.883 |
| 2-6 | 0.888 | 1.091 | - | 1.643 | 1.888 | 0.693 | - | 1.568 | - | 1.319 | 2.971 | - | - | 1.676 |
| Robust Avg | **1.38** | **1.23** | **1.53** | **1.32** | **1.75** | **1.67** | **1.95** | **1.43** | **2.00** | **1.59** | **1.81** | **1.83** | **1.63** | **1.81** |
| Rob StDev | 0.56 | 0.21 | 0.40 | 0.35 | 0.41 | 0.25 | 0.24 | 0.35 | 0.17 | 0.47 | 0.89 | 0.56 | 0.39 | 0.94 |
| Uncertainty | **0.23** | **0.10** | **0.19** | **0.15** | **0.21** | **0.10** | **0.11** | **0.15** | **0.08** | **0.20** | **0.45** | **0.25** | **0.17** | **0.42** |

Again, the scheme was used for both CP measurements and extrapolated value measurements from RS ($E_{LTE\_RS}$) and For SpanZero measurements ($E_{LTE\_SZ}$).

The goodness of the approach for determining the different consensus values, with relative standard deviation and uncertainty, is confirmed by the fact that the measure of the dispersion of the electric field values between all the measurement points (using 'standard deviation/Average' as a dispersion index) was significantly less than 10%, or, in general, the typical uncertainty of the measurements.

## 3. Results

### 3.1. Results of Characterisation Measurements

The results of the characterization measurements are shown, both in Table 4 and in subsequent graphs (Figures 4 and 5), directly compared to the values of the participants (results of the IC).

**Table 4.** Comparison of the results of characterization measurements (made in 2015 and 2016) and those of participants during IC (2016).

| "Sestet" | $E_{CP}$ (V/m) | | $E_{LTE\_RS}$ (V/m) | |
|---|---|---|---|---|
| | Characterization Measurements 2015/16 | IC Results 2016 | Characterization Measurements 2015/16 | IC Results 2016 |
| A | 0.71 | 0.77 | 1.60 | 1.64 |
| B | 0.80 | 0.77 | 1.54 | 1.65 |
| C | 0.67 | 0.76 | 1.60 | 1.59 |
| D | 0.80 | 0.77 | 1.59 | 1.69 |
| E | 0.69 | 0.73 | 1.55 | 1.56 |
| F | 0.78 | 0.75 | 1.61 | 1.60 |
| G | 0.75 | 0.71 | 1.56 | 1.56 |
| H | 0.72 | 0.79 | 1.62 | 1.70 |
| I | 0.78 | 0.75 | 1.60 | 1.69 |
| L | 0.73 | 0.82 | 1.55 | 1.66 |
| M | 0.74 | 0.74 | 1.61 | 1.66 |
| N | 0.78 | 0.83 | 1.65 | 1.72 |
| O | 0.68 | 0.75 | 1.60 | 1.59 |
| P | 0.72 | 0.77 | 1.53 | 1.58 |
| **Avg** | **0.74** | **0.77** | **1.60** | **1.63** |
| **St Dev %** | **6.0** | **4.2** | **2.1** | **3.3** |

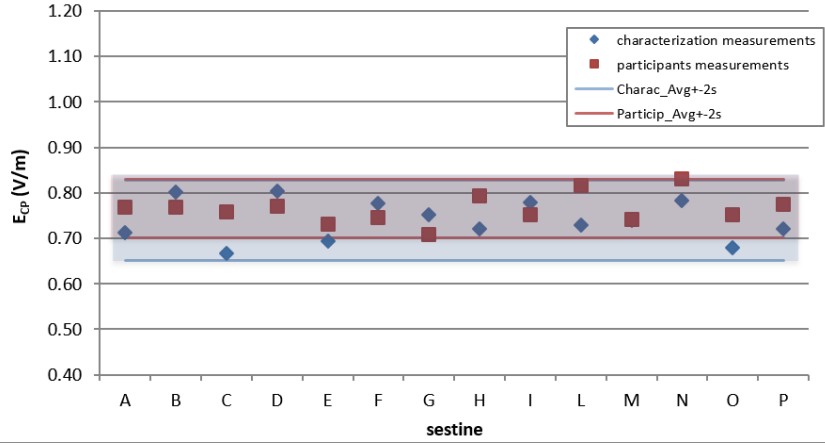

**Figure 4.** Graphical representation of the results of characterization measurements (sestet) and those of participants in CP mode with their variability ranges.

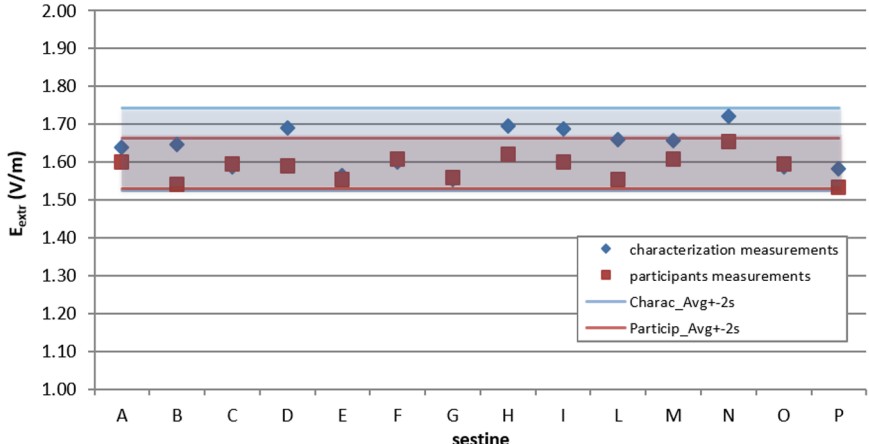

**Figure 5.** Graphical representation of the results of characterization measurements (sestet) of $E_{LTE\_RS}$ and participants' measurements with their variability ranges.

The value of each "sestet" is determined as the average of the values of the six positions. For characterization measurements, the value of each position is the average of the measurements of the two campaigns, while for the participants' measurements, the value of each position is the robust average of the results of the analysis per position (values of Tables 3 and 4).

The two graphs clearly show the good agreement between the characterization measurements and the intercomparison data. The greater dispersion of the characterization values respect to the participants results is probably due to the fact that the duration of the measurement campaign was longer than the measurement sessions of the participants.

*3.2. Intercomparison Results: Position Analysis*

The results of the participants' measurements, grouped by positions, are shown in the following Tables 3 and 4, along with the resulting averages. These values were used, as well as for comparison with characterization measurements, to determine the z-score of each measurement.

The single z-score is defined as:

$$z\text{-score} = (x_{i,k} - x_{m,k})/(\text{corrected StDev})_k$$

where $x_{i,k}$ is the measurement of the i-th participant in the k-th position, $x_{m,k}$ is the robust mean of the k-th position, (corrected StDev)$_k$ is the robust standard deviation of the k-th position.

Each sample of results was also represented by PomPlots [5].

The PomPlot displays the relative deviations (Deviation (D)/Median Absolute Deviation (MAD)) of the individual results ($x_i$) from the reference value (the Consensus value $x_m$, in this case) on the horizontal axis and relative uncertainties (u/MAD) on the vertical axis. For both axes, the variables are expressed as multiples of MAD, which is defined as the median absolute deviation from the reference value (the Consensus value $x_m$) [6].

It was decided to adopt this representation because each represented point directly includes both the uncertainty of the data itself and that of the consensus value.

The PomPlot is obtained by mapping the following combination:

$$((x_i - x_m)/MAD; u_{tot}/MAD) \text{ for each measurement of a specific participant}$$

where:

- $x_i$ is the single measurement of the participant,
- $x_m$ is the consensus value (robust average),
- MAD is the median absolute deviation from the Consensus value,
- $u_{tot}$ is the combined uncertainty obtained from the contributions of the participant's instrument and that of the consensus value.

On the following page, the PomPlots for the representation of some data samples for some positions are reported: 3 for measurements taken in CP mode and one for measurements of $E_{LTE\_RS}$.

In Figure 6, the graph (a) (position 2) shows a situation of good agreement: the values, all with similar uncertainty, are evenly distributed in the "min-max" variability range; graph (b) (position 1) shows a "mixed" situation in which there are several very concurring results (thickening of points in the center), a couple dispersed and one at the limit of acceptability, all still with similar uncertainty. Graph (c) (position 5) still shows a "mixed" situation in which; however, most of the data have an excellent degree of agreement except two that, despite the higher uncertainty, do not fall into the area of acceptability.

Finally, Figure 6d (position 10, Measurements of $E_{LTE\_RS}$) shows a rather atypical situation of two "thickenings" of points (apart from a value at the limit of acceptability), corresponding to the fact that the participants actually achieved all measurements around the two values of 1.3 V/m and 1.9 V/m alternated in the various shifts without this being attributed to a significant temporal variation.

*3.3. Intercomparison Results: Analysis by 'Sestet'*

In the analysis by 'sestet', as mentioned earlier, the results of the measurements taken by each participant in the six positions were treated as "measurement repetitions" and averaged (arithmetic mean); the resulting mean is used as a representative value of the measurements of the participant (sestet): the associated uncertainty is the one provided by the participant. Robust statistics was applied to the sample of these sestets to determine the consensus value to be assigned to the 'measurand' and its standard deviation and uncertainty. This consensus value, one for each measurement mode, is assumed as a representative value of magnitude ($E_{CP}$, $E_{LTE\_RS}$, $E_{LTE\_SZ}$).

The following Table 5 summarizes the results of the intercomparison.

Figure 7a–c shows the PomPlots representing the three data samples (one for each measurement mode) from which the values assigned to the sizes (measured) $E_{CP}$. $E_{LTE\_RS}$. $E_{LTE\_SZ}$ and the overall z-scores of the participants were determined (one for each measurement mode).

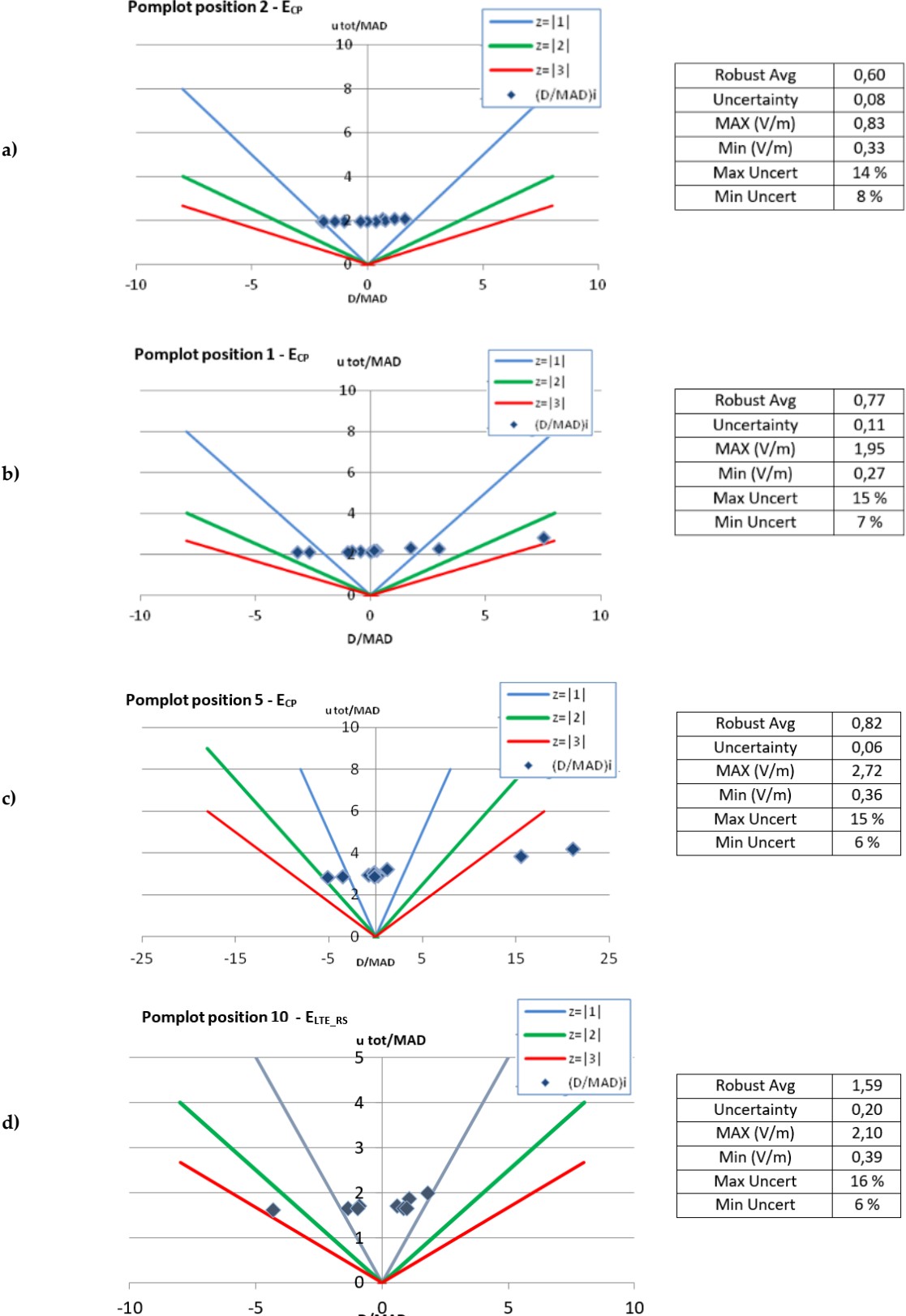

**Figure 6.** Graphical representation (PompPlots) of the results obtained in some positions: (**a**) $E_{CP}$ in position 2, (**b**) $E_{CP}$ in position 1, (**c**) $E_{CP}$ in position 5, (**d**) $E_{LTE\_RS}$ in position 10.

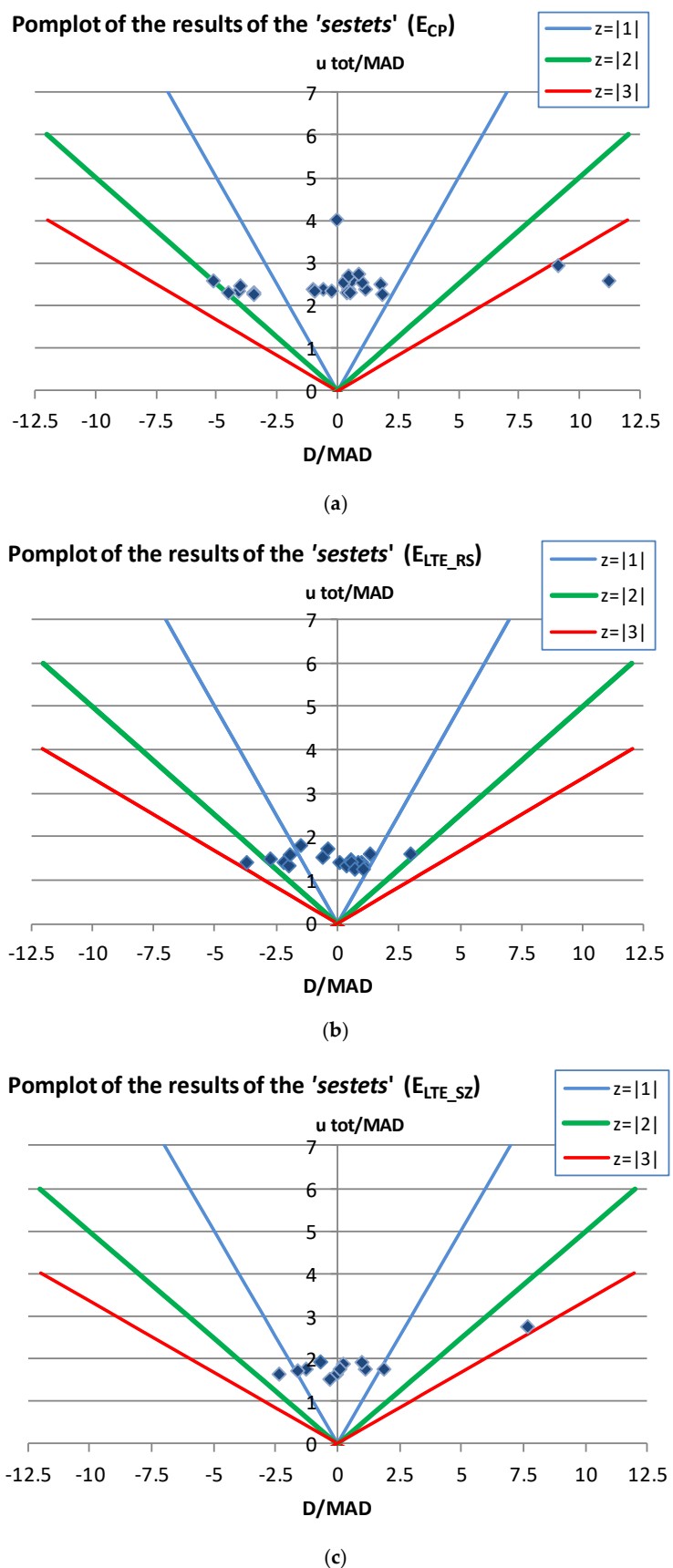

**Figure 7.** PompPlots of the results of the 'sestets' in the three modalities: (**a**) $E_{CP}$. (**b**) $E_{LTE\_RS}$. (**c**) $E_{LTE\_SZ}$.

**Table 5.** Results of sestets and final z-score for each measurement mode ($E_{CP}$, $E_{LTE\_RS}$, $E_{LTE\_SZ}$).

| Code of Participant | $E_{CP}$ | | $E_{LTE\_RS}$ | | $E_{LTE\_SZ}$ | |
|---|---|---|---|---|---|---|
| | $E_{CPi}$ (V/m) | z-Score | $E_{LTE\_RSi}$ (V/m) | z-Score | $E_{LTE\_SZi}$ (V/m) | z-Score |
| A01 | 0.69 | −0.54 | 1.81 | 0.43 | | |
| A02 | 0.82 | 0.25 | 1.92 | 0.80 | | |
| B03 | 0.73 | −0.32 | 1.70 | 0.07 | 2.35 | 0.68 |
| A04 | 0.82 | 0.23 | 1.93 | 0.82 | | |
| B05 | 1.58 | 5.03 | | | | |
| A06 | 0.83 | 0.33 | 1.15 | −1.76 | 1.81 | −0.83 |
| A07 | 0.42 | −2.24 | | | | |
| A08 | 0.83 | 0.30 | 1.89 | 0.70 | 2.08 | −0.06 |
| B09 | 0.48 | −1.88 | 1.19 | −1.63 | | |
| A10 | 0.39 | −2.46 | 0.78 | −3.00 | | |
| B11 | 0.43 | −2.18 | 1.02 | −2.21 | | |
| A12 | X | X | 1.54 | −0.48 | | |
| B13 | X | X | 1.58 | −0.34 | | |
| A14 | 0.33 | −2.80 | 1.31 | −1.23 | 1.56 | −1.53 |
| B15 | 0.70 | −0.49 | 1.76 | 0.28 | 2.03 | −0.22 |
| A16 | X | X | | | 2.53 | 1.18 |
| A17 | 0.88 | 0.65 | | | | |
| B18 | 0.80 | 0.14 | 2.40 | 2.41 | | |
| A19 | 0.87 | 0.56 | 1.82 | 0.46 | | |
| A20 | 0.83 | 0.29 | | | 3.86 | 4.91 |
| B21 | 0.78 | −0.02 | | | | |
| A22 | 0.94 | 1.01 | 1.99 | 1.04 | 1.73 | −1.07 |
| A24 | 1.77 | 6.17 | 1.21 | −1.55 | 2.15 | 0.12 |
| A25 | 0.76 | −0.13 | 1.85 | 0.58 | 2.32 | 0.60 |
| B26 | 0.86 | 0.51 | 1.94 | 0.86 | | |
| B27 | 0.48 | −1.86 | | | 2.12 | 0.04 |
| A28 | 0.94 | 1.02 | | | 1.95 | −0.43 |
| Rob Avg | 0.78 | | 1.68 | | 2.11 | |
| Rob StDev | 0.16 | | 0.30 | | 0.36 | |

Figure 8 shows the significant difference between the values of the electric field measured according to the three different modes.

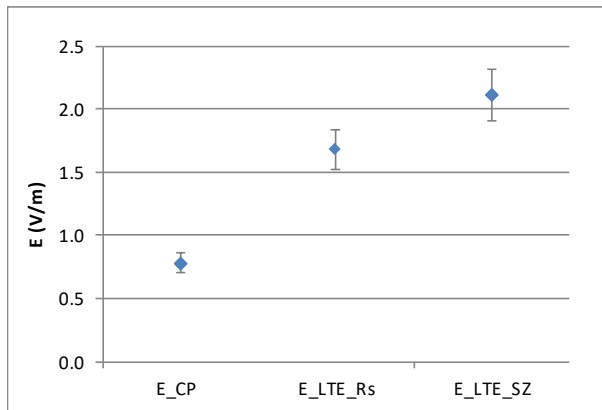

**Figure 8.** Comparison between the final results of the three modalities.

The E_CP value represents the field level detected during the measurement period. while the E_LTE_RS and E_LTE_SZ values are the maximum field level achievable in the measurement area. due to the BTS maximum emitted power. The results in Figure 8 show that. correctly. the first value is below the two maximum values. These last ones should be comparable: the difference within the two is consistent with the fact that the CEI 211-7E guide allows to use the Span Zero method only in order to exclude that the limit could be exceeded (and cannot be used to demonstrate effective exceeding of a limit).

Finally, Figure 9 shows the distribution of z-scores.

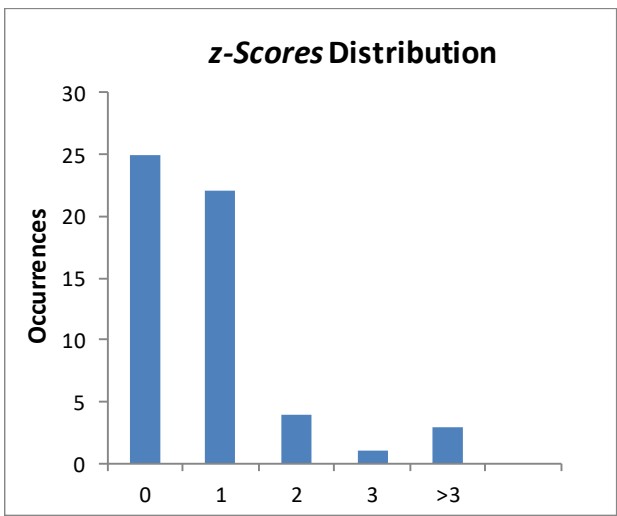

**Figure 9.** Distribution of all z-scores (by position analysis and by 'sestet' analysis) obtained by all the participants.

## 4. Conclusions

In this paper. the authors present the results of an intercomparison of electromagnetic field (EMF) measurements participated by expert laboratories (each one is the competent authority for the monitoring of its territory). The measurand was the EM field emitted by the source in actual operating conditions; the source was not controlled. For this reason. the authors considered important to describe the methodology implemented and used. The results are absolutely satisfactory (the worst Standard Deviation obtained in the 'position Analysis' is about 40%): most of the z-scores obtained by participants are 0 and 1.

**Author Contributions:** Conceptualization, L.A. (Lucia Ardoino), S.A. and L.A. (Laura Anglesio); Data curation, L.A. (Lucia Ardoino) and E.B.; Formal analysis, L.A. (Lucia Ardoino), S.A. and E.B.; Methodology, S.A. and L.A. (Laura Anglesio); Supervision, L.A. (Laura Anglesio); Writing—original draft, L.A. (Lucia Ardoino) and S.A. All authors have read and agreed to the published version of the manuscript.

**Funding:** This research received no external funding.

**Acknowledgments:** The authors thank all the participants. In addition, the authors would like to particularly thank Sabrina Barbizzi, from the National Centre for the National Laboratories Network (CN-LAB), Metrology Unit, ISPRA, for her support and contribution to the verification of the methodologies of statistical analysis.

**Conflicts of Interest:** The authors declare no conflict of interest.

## References and Note

1. Bienkowski, P. Interlaboratory comparisons in EMF survey measurements. *Environmentalist* **2009**, *29*, 130–134. [CrossRef]
2. ISO. *ISO/IEC 17043: Conformity Assessment: General Requirements for Proficiency Testing*; International Organization for Standardization: Geneva, Switzerland, 2010.
3. Technical Norm of the Italian Electrical Committee CEI 211-7/E "Guide for the measurement and the evaluation of electromagnetic fields in the frequency range 10 kHz–300 GHz, with reference to the human exposure". Annex E: Measurement of the electromagnetic fields from Base Radio Station for mobile telecommunication systems (2G, 3G, 4G). 2013.
4. Ardoino, L.; Barbieri, E.; Barbizzi, S.; Anglesio, L.; D'Amore, G.; Silvi, A.M. Confronti Interlaboratorio per misure di Campi Elettromagnetici: Analisi dei fattori d'influenza attraverso l'elaborazione statistica dei risultati del Circuito Ispra Ic015. *Boll. AIRP* **2012**, *171*, 149–156.

5.  Spasova, Y.; Pommé, S.; Watjen, U. Visualisation of interlaboratory comparison results in PomPlots. *Accredit. Qual. Assur.* **2007**, *12*, 623–627. [CrossRef]

6.  ISO. *ISO 13528: Statistical Methods for Use in Proficiency Testing by Interlaboratory Comparisons*; International Organization for Standardization: Geneva, Switzerland, 2005.

