# Peer review of "Selective Electromagnetic Measurements of 4G Signals: Results of an Italian National Intercomparison"

_environments, doi:10.3390/environments7010005_

Round 1

Reviewer 1 Report

The paper reports of an interlaboratory comparison of radiated field measurements at a location illuminated by a base station antenna for the purpose of quality assurance of the results provided by Italian National Environmental Agency.

The content is interesting and relevant to the scope of the journal.

English sometimes is so poor that the information conveyed is, in some cases, compromised. See in particular sentences in lines 56-58, 76, 78 (accessions?), 91 (what are individual evaluation cards?), 101 (measures? perhaps measurements), 101 (measurement modalities? perhaps receiver settings and measurement procedure?), 106 ("defining" two areas ... what is the issue that you are describing here?), 121 (divide the area in two? in two what?), the legend of the plots in Fig. 2 is in Italian, please use English everywhere, 275-278 (the sentence is quite involved).

Some technical comments.

Lines 145-147: Please explain what is the change that can be considered significant and which is the quantity measured by the "tool" for continuous narrow band capture.

150: "data provided with the cards" ... do you mean "raw manual transcription?" or similar?

165-169: please elaborate, this sentence is not clear. Why do you test for normality if you use robust statistics? Please provide a reference for Huber test. Further, again, why do you need to identify outliers if you are using Algorithm A in Annex C of ISO 13528? Do you mean outliers (that should not be excluded if you use robust statistics) or mistakes (to be excluded if identified)?

196-199: please clarify why you decided to use ordinary average and standard deviation for some sets of measured values and robust average and robust standard deviation for other ones.

Please add a conclusion section.

Reviewer 2 Report

The paper presents the results of an extensive work, and it covers the significant area of laboratory inter-comparisons at national level. The results were gained after relevant statistical computations and are of great significance for the EMF exposimetry community. However, methodological aspects are not sufficiently detailed, which is very important, and also, conclusions are lacking. More references are mostly welcome, including a discussion focused on similar approaches elsewhere. The originality is not well enough highlighted, even if it is present in the paper!

Bellow, please find our specific remarks, to improve the content.

General suggestions:

1. Paper title:  please remove the technical code name (IC_ISPRA2016_) and give a more generic title to the paper (like, eg. "Results of an Italian National Intercomparison" ; it is suggested to also underline that only downlink signals were assessed, maybe using this word in the title.

2. Along the paper (including abstract) there are lots of situations when the word "measures" was used, but the word "measurements" was the proper one; please apply corrections where necessary.

3. Along the paper, please first give the whole name/expression and introduce the abbreviation in the brackets, and only afterwards use directly the abbreviation.

4. The section of Conclusions is missing.

5. It is advised to insert much more references.

6. In the section "Materials & methods" it is necessary to present in detail the equipment used, the measurement procedure and the settings applied at each instrument. These can be done in a tabular form.

Abstract:

Line 16: instead of "implemented", you may insert "conducted";

Line 17: instead of "LTE mobile phone signals" better LTE mobile communications", since "phone" may conduct to the idea of measuring the uplink band contribution;

Line 21: please remove the code name of the project, IC_ISPRA2016_LTE and use a generic word.

Introduction:

Line 36: please remove "with";

Line 41-42: "intentionally variable over time" - this is not clear;

Inside this paragraph please provide some references for similar intercomparison studies. Also, please comment in two phrases about the characteristics of LTE signals and the proper instruments and sensors to be used in the measurements (narrow band).

Materials and methods:

Line 65: "in" should be "on";

Line 68: Where the measurements made in the far-field only?  Please give more details on the position of the network of measurement points with regard to its position to the source (in lambda-distances);

Line 71: please correct "Procedures" with "procedures";

Line 72: for "IC 2010" please provide a reference;

Line 74: "certificate LAT" - abbreviation not explained before;

Line 79: reference needed;

Line 86: "homogeneous areas" - maybe a proper expression can be found;

Line 96: it is not clear at this step if both bands were measured; were the measurements made in far-field conditions?...

Line 101: for each measurement mode, please present in a tabular mode the settings of the instruments used: RBW, VBW, SWT, Ch power bandwidth, etc. 

Fig. 1: on all figures please remove the space-sign!!

Fig. 2: in the legend, please complete "serie 2"

Line 136: for "Channel Power, Decoding RSi and SpanZero..." please insert abbreviation and then use them later in the text (see "CP" in line 138...);

Line 178: please revise the expression;

Line 182: "For" should be "for";

Results:

Line 190: in the title, it is also needed a change, from "measures" to "measurements";

Fig. 4 - in the legend, please correct "participants";

Fig. 5 - in the legend, please delete underline; correct to "participants";

Line 218:  insert the number of the relation; and also number the equations onward;

Line 263: Caption of Fig. 6 should be moved bellow the figure;  in Fig 6 (c) some series remained uncompleted in the legend;

Fig. 7 : caption should be moved bellow the figure; in one legend, the series remained uncompleted;

Conclusion section is missing!! Please synthesize the main outcomes of the measurement campaigns and compare them with similar ones.

Round 2

Reviewer 2 Report

The paper has been generally revised following the suggestions of the reviewer.

However, some small corrections still need to be made:

The title of the article, if still possible, can be changed to: "Selective Electromagnetic Measurements of 4G signals:....etc." 

The abbreviations were not always explained first time when they appeared or sometimes they were introduced later than when the whole words appeared first time in the text. These are true both for the body of the article and for the Abstract. Please take care at every abbreviation and apply all corrections.

Line 322 ends the relationship with the words "for each" - please complete "for each point" or similar.

Fig. 8 shows doubled abscissa.

I would kindly recommend not to insert Fig. 9 in the section Conclusions, but in the section above it, also with its description in the text. The comment may remain in the Conclusions, but a general one, not referring explicitly there to Fig. 9.

Author Response

Comment: The title of the article, if still possible, can be changed to: "Selective Electromagnetic Measurements of 4G signals:....etc."

Response: Title Changed

Comment: The abbreviations were not always explained first time when they appeared or sometimes they were introduced later than when the whole words appeared first time in the text. These are true both for the body of the article and for the Abstract. Please take care at every abbreviation and apply all corrections.

Response: I tried to correct every abbreviation, explaining them first time they appear in the text

Comment: Line 322 ends the relationship with the words "for each" - please complete "for each point" or similar.

Response: corrected on line 266

Comment: Fig. 8 shows doubled abscissa.

Response: sorry, I cannot see the doubled abscissa. On my screen, it seems a simple one.

Comment: I would kindly recommend not to insert Fig. 9 in the section Conclusions, but in the section above it, also with its description in the text. The comment may remain in the Conclusions, but a general one, not referring explicitly there to Fig. 9.

Response: text reviewed according to the suggestions.